# Clinical Evaluation of CA72-4 for Screening Gastric Cancer in a Healthy Population: A Multicenter Retrospective Study

**DOI:** 10.3390/cancers11050733

**Published:** 2019-05-27

**Authors:** Ping-Jen Hu, Ming-Yao Chen, Ming-Shun Wu, Ying-Chin Lin, Ping-Hsiao Shih, Chih-Ho Lai, Hwai-Jeng Lin

**Affiliations:** 1Division of Gastroenterology and Hepatology, Department of Internal Medicine, Mackey Memorial Hospital, Taitung 95054, Taiwan; a801891@hotmail.com; 2Division of Gastroenterology and Hepatology, Department of Internal Medicine, Shuang-Ho Hospital, New Taipei 23562, Taiwan; u90223@tmu.edu.tw; 3Division of Gastroenterology and Hepatology, Department of Internal Medicine, School of Medicine, College of Medicine, Taipei Medical University, Taipei 11031, Taiwan; vw1017@yahoo.com.tw; 4Division of Gastroenterology, Department of Internal Medicine, Wan-Fang Hospital, Taipei 11696, Taiwan; 5Department of Family Medicine, School of Medicine, College of Medicine, Taipei Medical University, Taipei 11031, Taiwan; greening1990@gmail.com; 6Center for Cell Therapy, Department of Medical Research, School of Medicine, China Medical University and Hospital, Taichung 40447, Taiwan; phs1027@gmail.com; 7Department of Microbiology and Immunology, Molecular Infectious Disease Research Center, Chang Gung University and Chang Gung Memorial Hospital, Linkou 33305, Taiwan

**Keywords:** CA72-4, esophagogastroduodenoscopy, gastric cancer, *Helicobacter pylori*, tumor marker

## Abstract

Early detection is important for improving the survival rate of patients with gastric cancer (GC). Serum tumor markers have been widely used for detecting GC. However, their clinical values remain controversial. This study aims to investigate the role of serum cancer antigen 72-4 (CA72-4) in the diagnosis of GC in a healthy population. A total of 7757 adults who underwent upper gastrointestinal endoscopy and serum CA72-4 level measurement in multicenters in Taiwan from January 2006 to August 2016 were recruited in this retrospective study. Risk factors for GC, serum tumor markers, and esophagogastroduodenoscopy (EGD) findings were evaluated. High serum levels of CA72-4 were found in 7.2% of healthy adults. CA72-4 level showed lower sensitivity (33.3%) but higher specificity (92.8%); however, the positive predictive value was quite low (0.18%). After adjustment of clinical risk factors for GC using EGD findings, gastric ulcer (adjusted odds ratio (aOR) = 2.11), gastric polyps (aOR = 1.42), and atrophic gastritis (aOR = 1.27) were significantly associated with high serum CA72-4 levels. Furthermore, both age (OR = 1.01) and *Helicobacter pylori* infection (OR = 1.44) exhibited a significant association with high serum CA72-4 levels. These results indicate that routine screening of CA72-4 levels for diagnosing GC in asymptomatic patients may be ineffective due to low sensitivity and low positive predictive value. The clinical utility of EGD findings along with serum CA72-4 level for screening healthy individuals with GC is warranted.

## 1. Introduction

Gastric cancer (GC) is one of the most common malignancies and is a major cause of cancer-related deaths worldwide [1]. The incidence of GC has been declining in the past few decades due to improvements in hygiene practices and effective eradication of *Helicobacter pylori* [1,2]. However, the survival rate of GC patients remains low, mostly due to delayed diagnosis [2]. The 5-year survival rate in Japan is much better than that in the western world, which is attributed to annual endoscopic screening for persons aged over 40 years and consecutive early tumor resections [3].

Although endoscopic examination and biopsy are the primary tools for diagnosing GC, high cost and invasiveness prevents them from being used as routine screening methods for asymptomatic persons, especially in low-incidence regions [4]. Several tumor markers have been used to assess patients with GC, including carcinoembryonic antigen (CEA), cancer antigen 19-9 (CA19-9), and CA72-4. Several studies indicated that a combination of these markers is useful in the diagnosis and prognosis of patients with GC [5,6].

CA72-4 was first described in the early 1980s as a novel antigen reactive to murine antibodies produced by mice that were immunized with membrane-enriched fractions of human metastatic mammary cancer cells [5,7,8]. It may be detected in patients with breast, colon, ovary, pancreatic, and gastric cancers [8]. Previous studies have reported that CA72-4 level was elevated in 29.9% (829/2774) of patients with GC and increased as the disease progressed [6]. However, its role as a screening marker for GS in healthy individuals is still under investigation. This study aims to assess serum CA72-4 as a screening tool for GC in asymptomatic populations.

## 2. Materials and Methods

### 2.1. Study Design and Population Collection

From January 2006 to August 2016, 7757 adults who had routine physical examinations, completed a self-administered questionnaire, and underwent esophagogastroduodenoscopy (EGD) [9] and serum CA72-4 measurement in multicenters, including Health Promotion Centers of Taipei Medical University Hospital, Wan Fang Hospital, Shuang Ho Hospital, and Everlife Health Services in Taiwan, were enrolled in this study. Demographic factors of the healthy individuals, including age, gender, weight and height (body mass index), smoking history, *H. pylori* infection, alpha-fetoprotein (AFP), CA72-4, CA19-9, and CEA, were recorded. The study participants were divided into abnormal group (CA72-4 > 6.9 ng/mL, *n* = 557) and normal group (CA72.4 ≤ 6.9 ng/mL, *n* = 7200) as suggested by Roche Diagnostics GmbH (Mannheim, Germany). The clinical risk factors for GC, other tumor markers, and EGD findings between the two groups were assessed. This study was approved by the Institutional Review Board of Taipei Medical University (no: N201702016). After providing a complete explanation of the study, a written informed consent was obtained from all participants. All clinical and biological samples were collected after obtaining patients’ consent. All the study methods were in accordance with the guidelines approved by the joint institutional review board and aforementioned governmental regulations.

### 2.2. Analysis of Serum Tumor Markers

Blood was obtained from all participants after fasting for 8 h. The CA72-4 level was assessed using a chemiluminescence kit (Roche Diagnostics GmbH). Serum CEA, CA19-9, and AFP levels were respectively measured using chemiluminescence assays (Beckman Coulter, Fullerton, CA, USA). The cutoff values for serum CEA, CA19-9, AFP, and CA72-4 were 5 ng/mL, 37 U/mL, 20 ng/mL, and 6.9 ng/mL, respectively, and were measured according to the manufacturer’s instructions.

### 2.3. Evaluation of EGD

The endoscopic examination was carried out using standard, forward-viewing video endoscopes by experienced gastroenterologists. The diagnosis and indications for biopsy were determined according to the American Society for Gastrointestinal Endoscopy guidelines [9]. The findings on EGD were classified as normal, benign (including reflux esophagitis, gastric ulcer, gastric polyp, gastric erosion, atrophic gastritis, and gastric submucosal tumor), or malignant (including esophageal cancer, gastric lymphoma, and GC) [9]. Each patient may have more than one EGD findings. All biopsy samples were evaluated by experienced pathologists, and the gastroenterologists applied the pathological report and made the definite EGD diagnosis.

### 2.4. Statistical Analysis

Categorical variables were compared with Pearson’s chi-square (χ^2^) test, and continuous variables were evaluated by Student’s *t*-test. Correlations between clinical risk factors for GC and CA72-4 level were further assessed by using multivariate logistic regression analyses. The logistic regression model was used to calculate odds ratios (OR) between both groups. Additional adjustments on clinical risk factors of GC, including age, gender, body mass index (BMI), smoking history, and *H. pylori* infection status were also carried out. All statistics analyses were performed by using SPSS statistical package (version 24.0, SPSS Inc., Chicago, IL, USA). A *p*-value of less than 0.05 was considered statistically significant.

## 3. Results

### 3.1. Demographic Characteristics of the Study Population

A total of 7757 adults who underwent health examinations from January 2006 to August 2016 in various health promotion centers in Taiwan were recruited in this study (Figure 1). The clinical profiles of the enrolled patients are shown in Table 1. The mean age was 45.6 ± 11.1 years (mean ± SD, range: 20–82 years); majority of the patients were men (*n* = 4704, 60.6%). A total of 1668 patients (21.9%) had a smoking history, and 1611 (21.1%) had *H. pylori* infection.

### 3.2. Measuring Serum CA72-4 in Healthy Population

About 557 patients had elevated CA72-4 levels (7.2%). As shown in Table 2, 181 patients (2.3%), had high serum CEA levels, while 52 (0.87%) had high serum CA19-9 levels. In the further analyses, three patients had GC based on the results of endoscopic examination and pathological findings. The CA72-4 levels of these three patients were 1.46, 1.69, and 7.47 ng/mL, respectively. In this study, only one patient had abnormal CEA, CA19-9, and CA72-4 levels and was found to have benign gastric disease.

The mean value of CA72-4 values in high and normal CA72-4 groups were 14.2 ± 7.1 and 2.0 ± 1.3 ng/mL, respectively (*p* < 0.001). Our results showed that older age (*p* = 0.007) and presence of *H. pylori* infection (*p* < 0.001) presented in the high CA72-4 group. Age, male, smoking history, BMI, and *H. pylori* infection were further analyzed using a multivariate regression model. As shown in Table 3, both age (OR = 1.01; 95% confidence interval (CI) = 1.00–1.02) and *H. pylori* infection (OR = 1.44; 95% CI = 1.19–1.76) had significant association with high serum CA72-4 level.

### 3.3. Association between Serum CA72-4 and EGD Findings

We then correlated EGD findings with CA72-4 level. As shown in Table 4, subjects with gastric ulcer (*p* < 0.001), gastric polyps (*p* = 0.003), and atrophic gastritis (*p* = 0.047) tend to have high serum CA72-4 level. Univariate and multivariate binary logistic regression analyses were then performed to evaluate their correlations. Univariate analysis showed similar outcomes: gastric ulcer (OR = 2.20; CI = 1.63–3.99), gastric polyps (OR = 1.45; CI = 1.14–1.85), and atrophic gastritis (OR = 1.34; CI = 1.00–1.78). After adjusting by age, male, smoking history, BMI, and *H. pylori* infection, multivariate logistic regression revealed that gastric ulcer (adjusted OR (aOR) = 2.11; CI = 1.56–2.86) and gastric polyps (aOR = 1.42; CI = 1.11–1.81) possessed significant positive correlations with high CA72-4 level, while atrophic gastritis (aOR = 1.27; CI = 0.95–1.69) exhibited a negative correlation (Table 5).

### 3.4. Association between CA72-4 Serum Level and GC

Among 7757 enrolled objects, only 3 persons had GC as confirmed by EGD and pathology. One patient had a high CA72-4 level, with sensitivity of 33.3% and specificity of 92.8%. This finding indicated that CA72-4 had a very low positive predictive value (0.18%) of detecting GC in healthy population. In addition, the negative predictive value was 99.97%, and the total diagnostic accuracy rate was 92.8%.

## 4. Discussion

There are several important findings that may be helpful in clinical interpretation. Our results indicated that routine screening of CA72-4 for GC in asymptomatic population seems unnecessary due to extremely low positive predictive value (0.18%). High CA72-4 value is correlated with gastric ulcers, gastric polyps, and atrophic gastritis, but not with GC. In addition, old age, *H. pylori* infection, and high CEA value are positively associated with high CA72-4 level.

CA72-4 is commonly regarded as one of the most specific and sensitive markers for monitoring GC [6,10]. A systemic review performed by the Task Force of the Japanese Gastric Cancer Association in 2014 revealed that the overall positive rate of CA72-4 for GC was 16–70%, which makes it the most highly correlated serum tumor marker for GC, although the sensitivity remained low [6]. Liang reported that CA 72-4 was positive in 27.6% patients with GC, which was higher than that in the control (14.8%, *p* < 0.001) [11]. In the present study, we enrolled much more healthy subjects (*n* = 7757) than the previous studies and found a lower positive rate of CA72-4 (7.2%) in the general population.

The sensitivity and specificity of CA72-4 for GC were significantly higher in patients with advanced stage GC than in those with early stage GC [6,12]. In a retrospective study in Turkey, CA72-4 were detected in 82% of GC patients with liver metastasis [13]. Elevated CA72-4 level was associated with tumor stages, tumor depth, nodal involvement, and metastasis [14,15,16]. In a review study, Shimada found that CA72-4 positive rate increased as the disease progressed (stage I: 12%, stage II: 15.6%, stage III: 36.7%, and stage IV: 49.6%) [6]. This is because the CA72-4 antigen in cancer cells is not released into the circulation in the early stage and may enter the circulation as the lymph vessels or veins are invaded with cancer cells [17]. Together, CA 72-4 seems to be useful in assessing the extent of cancer invasion before operation; however, it is not qualified as a screening tool for GC in a healthy population.

Most of the study objects were limited to GC patients and indicated that CA72-4 may be helpful in disease monitoring and prognosis [6,8,12,18]. A meta-analysis conducted in Chinese population revealed that odds ratio of CA72-4 for GC was as high as 32.9 [18]. However, the value of CA72-4 in predicting GC in asymptomatic population remained unclear. Noticeably, the enrolled subjects were limited to patients with GC in the previous studies and the positive prediction rate could not be calculated accurately. Our study enrolled patients from multicenters in Taiwan. In this study, the total diagnostic accuracy rate was 92.8%, but it was not considered significant due to the low positive predictive value of CA72-4 for GC (0.18%). This was the first multicenter large-scale study to evaluate the clinical value of CA72-4 for screening of GC in the asymptomatic healthy population.

It has been known that a combination of multiple tumor markers is more effective in evaluating GC than a single tumor marker [11,19,20]. A retrospective study revealed that the combined detection of CEA, CA19-9, CA242, and CA72-4 possessed a sensitivity of 82.6% and a specificity of 83.3% [21]. Yu and Zeng found that the establishment of a GC screening system based on serum levels of CEA, CA19-9, and CA72-4 biomarkers seemed to be an effective approach [19]. However, in their study, the sample size was very limited and may not reach a solid conclusion. In this study, only one person was found with triple positive of CEA, CA19-9, and CA72-4, but was not diagnosed with GC.

Our study indicated that high CA72-4 value is correlated with *H. pylori* infection. *H. pylori* infection has been considered as a leading cause of gastric cancer [22]. Epigenetic reprogramming of host cell genome is initiated by *H. pylori* infection through a direct microbe–gastric epithelial cell interaction, and possibly plays a key role in gastric carcinogenesis [23]. However, whether epigenetic modification in gastric cancer correlated with the CA72-4 level, has not been well studied. We also investigated the association between serum CA72-4 level and benign gastric disease. Chronic gastric diseases are usually independent of gastric cancer occurrence; therefore, these populations are potentially GC patients in the long run. Comprehensive large-scale prospective studies are necessary to clarify this issue.

In Taiwan, GC is the seventh leading cause of cancer-related deaths [24]. The economic burden of advanced GC accounts for 0.08% of the Taiwanese medical expense [25]. Although endoscopic biopsy along with histopathological evaluation is the gold standard for diagnosing GC, its invasiveness and high expense precludes it as a routine screening method. Establishing a noninvasive and cost-effective comprehensive screening system is the major goal of future studies.

Although this study enrolled a larger proportion of healthy individuals who underwent routine physical examination from multicenters, some limitations might exist, including potential confounders, histopathological examination, and various cancer types. First, GC may be possibly missed by endoscopic examination, either by misinterpretation or inadequate sampling [26]. Second, in addition to GC, CA72-4 is also related to breast, colon, and pancreatic cancers [8]. In this study, we did not determine the presence of other cancers and focused only on GC. There may be some patients who had cancers other than GC.

## 5. Conclusions

In this study, we demonstrated that high serum CA72-4 level is positively associated with older age, *H. pylori* infection status, gastric ulcer, gastric polyps, and atrophic gastritis. The total diagnostic accuracy rate of CA72-4 for detecting GC was 92.8%; however, the positive predictive value was low (0.18%). Routine screening of CA72-4 for GC in asymptomatic patients may be ineffective due to the low positive predictive rate. A combination of EGD findings and serum CA72-4 for screening healthy individuals with GC is recommended.

## Figures and Tables

**Figure 1 cancers-11-00733-f001:**
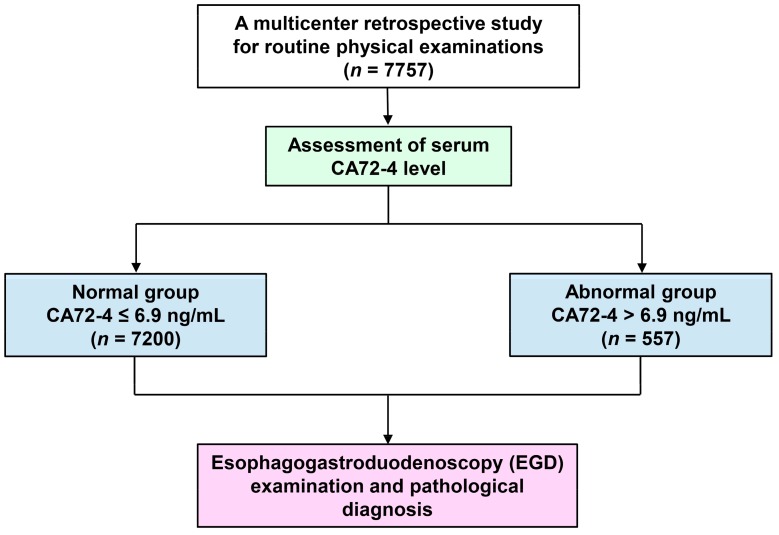
Flowchart of the population selection, identification, and analysis in a multicenter retrospective study.

**Table 1 cancers-11-00733-t001:** Demographic characteristics of patients who were recruited in this study.

Parameter	Number (%)
Total population	7757
Age (mean ± SD)	45.6 ± 11.1
Gender	
Male, *n* (%)	4704 (60.6%)
Female, *n* (%)	3053 (39.4%)
Total number	7757
Smoking history	
Yes, *n* (%)	1668 (21.9%)
No, *n* (%)	5950 (78.1%)
Total number	7618
BMI ^†^ (mean ± SD), kg/m^2^	23.8 ± 3.6
*H. pylori* infection	
Positive, *n* (%)	1611 (21.1%)
Negative, *n* (%)	6030 (78.9%)
Total number	7641
CA72-4 (mean ± SD), ng/mL	2.8 ± 3.9
Total number	7757
CA19-9 (mean ± SD), U/mL	10.3 ± 8.3
Total number CA19-9 positive, *n* (%)	5989 52 (0.87%)
CEA (mean ± SD), ng/mL	1.7 ± 1.2
Total number CEA positive, *n* (%)	7703 181 (2.35%)

^†^ BMI: Body mass index.

**Table 2 cancers-11-00733-t002:** Clinical risk factors of gastric cancer (GC) and tumor markers in high and normal CA72-4 level groups.

Variable	High CA72-4 Level (*n* = 557)	Normal CA72-4 Level (*n* = 7200)	*p*-Value ^†^
Age (years ± SD)	46.8 ± 12.2	45.5 ± 11.0	**0.007**
Male, *n* (%)	346 (62.1%)	4358 (60.5%)	0.459
Smoking history, %	22.2%	21.9%	0.848
BMI (mean ± SD), kg/m^2^	24.0 ± 3.6	23.8 ± 3.6	0.394
*H. pylori* infection, %	27.5%	20.6%	**<0.001**
CA72-4 (mean ± SD), ng/mL	14.2 ± 7.1	2.0 ± 1.3	**<0.001**
CA19-9 (mean ± SD), U/mL	10.1 ± 8.9	10.3 ± 8.3	0.744
CEA (mean ± SD), ng/mL	2.0 ± 1.2	1.7 ± 1.2	**<0.001**

^†^ Statistical significant difference is indicated by a bold number.

**Table 3 cancers-11-00733-t003:** Multivariate logistic regression analysis for independent predicators of high serum CA72-4 level.

Variable ^‡^	Multiple Analysis ^†^		
	*p*-Value ^¶^	OR	(95% CI)
Age (Years)	**0.014**	1.01	1.00–1.02
Male	0.476	1.07	0.89–1.29
Smoking history	0.911	0.99	0.80–1.22
BMI (kg/m^2^)	0.515	1.01	0.98–1.03
*H. pylori* infection	**<0.001**	1.44	1.19–1.76

^†^ OR: odds ratios; CI: confidence intervals; BMI: body mass index. ^‡^ Multivariate model included age, male, smoking history, BMI, and *H. pylori* infection. ^¶^ Statistical significant difference is indicated by a bold number.

**Table 4 cancers-11-00733-t004:** Difference in esophagogastroduodenoscopy (EGD) findings between high and normal CA72-4 level groups.

EGD Finding	High CA72-4 Level (*n* = 557)	Normal CA72-4 Level (*n* = 7200)	*p*-Value ^†^
Normal	214 (38.4%)	3454 (48%)	
Reflux esophagitis	143 (25.7%)	1870 (26%)	0.877
Gastric ulcer	53 (9.5%)	328 (4.6%)	**<0.001**
Gastric polyps	83 (14.9%)	776 (10.8%)	**0.003**
Gastric erosions	74 (13.3%)	792 (11%)	0.099
Atrophic gastritis	57 (10.2%)	566 (7.9%)	**0.047**
Gastric submucosal tumor	23 (4.1%)	291 (4.0%)	0.920
Esophageal cancer	0 (0%)	2 (0.03%)	0.694
Gastric cancer	1 (0.18%)	2 (0.03%)	0.079

^†^ Statistical significant difference is indicated by a bold number.

**Table 5 cancers-11-00733-t005:** Univariate and multivariate logistic regression of high serum CA72-4 level for different EGD findings.

Outcome of EGD Finding	Univariate ^†^		Multivariate ^‡^	
	OR	95% CI	Adjusted OR	Adjusted 95% CI
Reflux esophagitis	0.99	0.81–1.20	0.98	0.80–1.20
Gastric ulcer	2.20	1.63–2.99 ***	2.11	1.56–2.86 ***
Gastric polyps	1.45	1.14–1.85 **	1.42	1.11–1.81 **
Gastric erosions	1.24	0.96–1.60	1.13	0.87–1.47
Atrophic gastritis	1.34	1.00–1.78*	1.27	0.95–1.69
Gastric submucosal tumor	1.02	0.66–1.58	0.98	0.63–1.51
Gastric cancer	6.47	0.59–71.5	4.54	0.37–56.32

^†^ OR: odds ratios; CI: confidence intervals; BMI: body mass index. ^‡^ Multivariate model included age, male, smoking history, BMI, and *H. pylori* infection, and serum CA72-4 level (high or normal). * *p* < 0.05; ** *p* < 0.01; *** *p* < 0.001.

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
