# Peer review of "Clinical Evaluation of CA72-4 for Screening Gastric Cancer in a Healthy Population: A Multicenter Retrospective Study"

_cancers, 2019, doi:10.3390/cancers11050733_

Reviewer 1 Report

In this study, the author declare that routine screening of CA72-4 for GC in asymptomatic population seems unnecessary  due to extremely low positive predictive value. High CA72 - 4 value is correlated with gastric ulcers, gastric polyps, and atrophic gastritis but not with GC. In addition, old age, H. pylori infection, and high CEA value are positively associated with high CA72 - 4 level.

Overall the manuscript is original and well defined. The results are significant, appropriately interpreted and supported the conclusions. The study is correctly designed. were performed with the highest technical standards. The English language is appropriate.

However, I have a few concerns and related suggestions.

The authors did not discuss regarding the the long-term effects of H. pylori-induced epigenetic alterations which may occur during gastric carcinogenesis.

I suggest to briefly include this point in the discussion mentioning an interesting work in this field (Chiariotti et al., 2013).

Author Response

Response to Reviewer 1 Comments

Review #1

In this study, the author declare that routine screening of CA72-4 for GC in asymptomatic population seems unnecessary due to extremely low positive predictive value. High CA72 - 4 value is correlated with gastric ulcers, gastric polyps, and atrophic gastritis but not with GC. In addition, old age, H. pylori infection, and high CEA value are positively associated with high CA72 - 4 level.

Overall the manuscript is original and well defined. The results are significant, appropriately interpreted and supported the conclusions. The study is correctly designed. were performed with the highest technical standards. The English language is appropriate.

However, I have a few concerns and related suggestions.

The authors did not discuss regarding the the long-term effects of H. pylori-induced epigenetic alterations which may occur during gastric carcinogenesis.

I suggest to briefly include this point in the discussion mentioning an interesting work in this field (Chiariotti et al., 2013).

Author response: Thank you very much for the constructive comments. We have discussed the issue of epigenetic modification by H. pylori and cited the reference in the Discussion section. Please refer to page 6 (lines 204-208).

Reviewer 2 Report

It is a negative data and important data, too. These kinds of "markers" will not be useful in high endemic area and in the area endoscopy are prevailing.

The authors should just mention, gastric ulcer and other chronic gastric disease are usually not independent on gastric cancer occurrence; thus these population potentially be a GC patients in a long run. Larger scale of prospective study would be necessary to eliminate all the trials, but it would not be feasible.

Author Response

Response to Reviewer 2 Comments

Reviewer #2

It is a negative data and important data, too. These kinds of "markers" will not be useful in high endemic area and in the area endoscopy are prevailing.

The authors should just mention, gastric ulcer and other chronic gastric disease are usually not independent on gastric cancer occurrence; thus these population potentially be a GC patients in a long run. Larger scale of prospective study would be necessary to eliminate all the trials, but it would not be feasible.

Author response: We sincerely appreciate the insightful comments provided by the reviewer. We have addressed the concerns and incorporated this issue in the Discussion section. Please refer to page 6, lines 208-212.

Reviewer 3 Report

1) Did you use different levels of CA 72-4 to see if the positive predictive value can be increased?

2) Did you look at high risk population rather than screening healthy population?

Author Response

Response to Reviewer 3 Comments

Reviewer #3

1) Did you use different levels of CA 72-4 to see if the positive predictive value can be increased?

Author response: Thank you very much for the valuable suggestion. At the beginning of this study, we did use different cutoff values for CA72-4, but all cut points revealed an extremely low positive predictive value. We finally chose 6.9 ng/mL as the cutoff value according to the manufacturer’s instructions, and also because it was the most commonly used cutoff value in previous studies.

2) Did you look at high risk population rather than screening healthy population?

Author response: Most studies have been conducted on high-risk populations, including studies on gastric cancer and peptic ulcer diseases; however, there was no study on the healthy population. We therefore analyzed the correlation between CA72-4 value and risk factors for gastric cancer in healthy individuals, and the results indicated that old age and H. pylori infection are positively associated with high CA72-4 level. It was the first large-scale, multicenter study to evaluate the significance of the clinical value of CA72-4 levels in screening for gastric cancer in the asymptomatic healthy population. We have incorporated this issue in the Discussion section. Please refer to page 6, lines 188-193. We thank the reviewer for the valuable comments.
